# Design and Performance Verification of a Novel RCM Mechanism for a Minimally Invasive Surgical Robot

**DOI:** 10.3390/s23042361

**Published:** 2023-02-20

**Authors:** Hu Shi, Zhixin Liang, Boyang Zhang, Haitao Wang

**Affiliations:** School of Mechanical Engineering, Xi’an Jiaotong University, Xi’an 710049, China

**Keywords:** remote center of motion, minimally invasive surgery, robot, master–slave control, planar mechanism

## Abstract

Minimally invasive surgical robots have the advantages of high positioning accuracy, good stability, and flexible operation, which can effectively improve the quality of surgery and reduce the difficulty for doctors to operate. However, in order to realize the translation of the existing RCM mechanism, it is often necessary to add a mobile unit, which is often bulky and occupies most space above the patient’s body, thus causing interference to the operation. In this paper, a new type of planar RCM mechanism is proposed. Based on this mechanism, a 3-DOF robotic arm is designed, which can complete the required motion for surgery without adding a mobile unit. In this paper, the geometric model of the mechanism is first introduced, and the RCM point of the mechanism is proven during the motion process. Then, based on the establishment of the geometric model of the mechanism, a kinematics analysis of the mechanism is carried out. The singularity, the Jacobian matrix, and the kinematic performance of the mechanism are analyzed, and the working space of the mechanism is verified according to the kinematic equations. Finally, a prototype of the RCM mechanism was built, and its functionality was tested using a master–slave control strategy.

## 1. Introduction

Robot-assisted surgery has the advantages of high positioning accuracy, good stability, and flexible operation. During the movement of the minimally invasive surgical robot, it is necessary to ensure that the surgical instruments always pass through the fixed point of the spatial position of the incision and cannot pull the incision to avoid secondary injury to the patient. In order to ensure safety during surgery, some motion constraints are required, which is the key to designing minimally invasive surgical robots.

Generally, three methods are used to meet the motion constraints of surgical instruments, namely multi-joint linkage, passive joint constraint, and mechanism constraint. Among them, multi-joint linkage realizes the fixed point of the robotic arm during the movement process through kinematics and control algorithms, and the passive joint constraint can be affected by the patient’s breathing, heartbeat, and other physiological activities. Therefore, the above two methods were basically adopted in the early research, and currently mechanism constraint is mostly used to realize the motion constraints of minimally invasive surgical robotic arms.

The concept of the RCM (remote center of motion) mechanism was first proposed by Taylor et al., who also initially applied the mechanism to the field of the minimally invasive surgical robot [1,2]. If a part of or a point of the mechanism always passes through a fixed point far away from the mechanism itself (or confined in a narrow space) during the process of motion and there is no physical hinge constraint at that point, this kind of mechanism can be called an RCM mechanism [3]. The characteristics of this kind of mechanism can precisely meet the movement requirements of minimally invasive surgical instruments, so it is widely used in minimally invasive surgical mechanical arms. The RCM mechanism is the basis of the minimally invasive surgical robotic arms. Different RCM mechanism constructions result in different motion performances of the robotic arms, which have huge impacts on the surgical effects. Therefore, analysis and innovation related to the RCM mechanism are important research topics in the field of minimally invasive surgical robots. Common RCM mechanisms can be divided into spherical mechanisms, arc-shaped mechanisms, and double-parallelogram mechanisms. An arc-shaped mechanism is the simplest way to realize the principle of RCM. Ut-neu [4] is a minimally invasive surgical robot for neurosurgery that was established based on an arc-shaped mechanism. Yousef et al. [5] designed a crossed double-arc mechanism, which can realize two independent degrees of freedom (DOF) of yaw and pitch of the surgical instrument. Aiming at miniaturization, Peter et al. [6] designed a compact and modular minimally invasive surgical robotic arm based on an arc-shaped mechanism. Each robotic arm weighs less than 2 kg and can be fixed on the operating table for use, but the pitch range it can realize is only 35°.

Some other researchers have used spherical mechanisms to meet the motion constraints of minimally invasive surgical robotic arms. Lum et al. [7] divided spherical mechanisms into two types, the serial type and the parallel type, and an experimental evaluation showed that the serial spherical mechanism has the advantage of avoiding motion interference over the parallel type. A kinematics analysis and optimization were carried out using the serial spherical mechanism, and the optimal linkage length was obtained. Rosen et al. [8] established the Raven robot system for minimally invasive surgery based on the spherical mechanism.

The double-parallelogram mechanism is currently the most widely used RCM mechanism. For example, the famous Da Vinci surgical robot [9] is based on the double-parallelogram mechanism to achieve the remote center of motion. Tan [10] and Kong [11] proposed a new type of RCM mechanism based on the deformation of the double-parallelogram mechanism. The mechanism proposed by Kong can realize two DOF of yaw and translation of surgical instruments.

However, there is no RCM mechanism in the current research that can realize the translation of the surgical instrument directly, and an additional joint has to be employed at the end of the robotic arm. The above-mentioned arc mechanism, spherical mechanism, and double-parallelogram mechanism can only realize the yaw and pitch of surgical instruments, so it is necessary to adopt a prismatic joint at the end to perform the translation of surgical instruments. Unfortunately, due to the large volume of the prismatic joint, it may cause interference between different mechanical links during the movement. In order to address the issue, a novel planar 2-DOF RCM mechanism was proposed that can realize the translation of surgical instruments through the mechanism itself. Based on the mechanism, a robotic arm for minimally invasive surgery was developed, and the control performance was verified through an experiment.

## 2. Mechanism Design

### 2.1. Conceptualization

Surgical robots need to achieve multiple degrees of freedom of movement during surgery, but the DOF usually achieved by linkage mechanisms are the pitch and translation of the surgical instruments [12]. Therefore, in this paper, we design a novel planar RCM mechanism to achieve these two DOF of surgery simultaneously.

Analyzing the characteristics of the double-parallelogram mechanism in Figure 1, it is found that the virtual quadrilateral *ABCR* is the key to realizing the remote center of motion, and the virtual quadrilateral is guaranteed to be a parallelogram (*BC*//*AR* and *AB*//*CR*) by the parallelograms *AODE* and *BEFC*. In this parallelogram, the linkage *BC* performs a pure translational motion. Under the constraints of the two parallelograms, *AODE* and *BEFC*, the linkage *BC* is always parallel to the base during the movement process. Although it moves in the *x* and *y* directions, the two directions are coupled, so the DOF is still 1. Therefore, if the linkage *BC* can have two independent translational DOF, then the linkage *CR* can achieve two DOF in pitch and translation, and due to the constraint of the virtual parallelogram *ABCR*, the linkage *CR* will always pass through the virtual fixed point *R*. Considering that the parallelogram *BEFC* in the double-parallelogram mechanism is preserved, the parallelogram *AODE* can be replaced by two mobile joints, as shown in the lower left of Figure 1. In this mechanism, due to the existence of two mobile joints, the linkage *BC* has two independent translational DOF, so the end output linkage *CR* can achieve two DOF in pitch and translation. In order to reduce the volume of the mechanism, the positions of the two mobile joints can be changed, and the mechanism can be transformed into the construction shown in the lower right of Figure 1.

The mechanism can directly realize the two DOF of pitch and translation of the surgical instrument, and after adding a revolute joint whose axis passes through the fixed point to drive the entire mechanism to yaw, the yaw motion of the surgical instrument can be realized, as shown in Figure 2. The mechanism is derived from the double-parallelogram mechanism. On one hand, it inherits the advantage of occupying little space above the patient’s body. On the other hand, it can directly realize the pitch and translation of the surgical instrument without the need to add a mobile joint at the end, which can avoid the risk of interference in the movement of the manipulator to the maximum extent.

### 2.2. Mathematical Proof

As shown in Figure 3, two mobile joints of the mechanism are selected as active joints, and it is assumed that when both mobile joint I and mobile joint II are at the initial position, point *C* coincides with point *O*; that is, *d*_1_ = *d*_2_ = 0.

Due to the motion of mobile joint I and mobile joint II, the angle between the linkage *A*_2_*B*_2_ and the x-axis can be expressed as
(1)$$\theta {=\tan}^{-1}\frac{d_{2}}{d_{1}-a}$$
where *d*_1_ and *d*_2_ are the displacements of mobile joint I and mobile joint II, and *a* is the distance between point *A*_1_ and point *C*. The expression for the line *A*_2_*B*_2_ can be written as
(2)$$P=P_{A_{2}}+k\omega \;k\in R$$
where:

***P*** = [*x y*]^T^ is the coordinate of any point on the line;

***P***_*A*_2__ = [*d*_1_ + *b* − *a d*_2_]^T^ is the coordinate of the point *A*_2_;

*b* is the length of the linkage *A*_1_*A*_2_;

***ω*** = [cos*θ* sin*θ*]^T^ is the direction vector of the straight line *A*_2_*B*_2_.

Then, Formula (2) can be written as
(3)$$\frac{y-d_{2}}{x-\left(d_{1}+b-a\right)}=\frac{d_{2}}{d_{1}-a}$$

Obviously, Formula (3) is constant for the point *R*[*b* 0]^T^, so the straight line *A*_2_*B*_2_ always passes through the point *R*[*b* 0]^T^, regardless of the movement of mobile joint I and mobile joint II. Therefore, the mechanism is an RCM mechanism, which can achieve pitch and translation in a two-dimensional plane around the virtual fixed point *R*.

The motion output rod (linkage *A*_2_*B*_2_) of the mechanism is coupled with two DOF of rotation around the RCM point and the translation along the axis. In order to analyze the condition that the rod performs a pure rotation around the RCM point without translation, it can be considered that point *E* and point *A*_1_ make circular motions around the origin at this time, and the coordinates of *A*_1_ can be expressed as [*d*_1_ − *a d*_2_]^T^. Then
(4)$${\left(d_{1}-a\right)}^{2}+d_{2}^{2}=C$$
where *C* is a constant whose range satisfies 0 < *C* < *c*^2^ and *c* is the length of the linkage *A*_1_*E.* When the displacements *d*_1_ and *d*_2_ of mobile joint I and mobile joint II satisfy Formula (4), the output rod of the mechanism performs a pure rotation around the RCM point, and the corresponding insertion depth of the surgical instrument is $c-\sqrt{{\left(d_{1}-a\right)}^{2}+d_{2}^{2}}$.

In order to analyze the necessary conditions for the pure translational motion of the output rod along the axis, *θ* can be set as a constant. Then
(5)$$\frac{d_{2}}{d_{1}-a}=\tan\theta$$

The linkage *A*_2_*B*_2_ will always maintain an angle (*θ*) with the *x*-axis and be able to perform translation along its own axis.

### 2.3. Singularity and Kinematic Performance Analysis

It is known that parallelograms tend to encounter singularity when all links are collinear [13]. When it is in the singular position, the mechanism will lose one or more existing DOF. It is necessary to avoid the mechanism in the singular position in advance. The singularity of the mechanism can generally be found by setting the value of the determinant of the Jacobian matrix of the mechanism to 0. The Jacobian matrix (***J***) is used as a linear transformation to relate the Cartesian velocity of the end to the joint velocity using the following formula:(6)$$\left[\dot{\theta }]=J^{-1}(\theta )[v\right]$$

When the Jacobian matrix (***J***) is invertible, if the Cartesian velocity of the end is known, the joint velocity can be obtained using Formula (6). However, the Jacobian matrix is a function of the joint angle (*θ)*, which is not necessarily invertible for all joint angles in all pose configurations. If in a certain configuration, the value of the determinant of the Jacobian matrix is det (***J***) = 0; that is, the Jacobian matrix is irreversible. In this configuration, the mechanism cannot achieve a speed in a certain direction, which means it is in a singular situation.

In Figure 3, the closed-loop position vector equation of the mechanism is established based on the coordinate system *O-xy.*
(7)$$P_{D}=P_{A_{2}}-c\omega$$
where ***P***_*A*_2__ = [*d*_1_ + *b* − *a d*_2_]^T^ is the coordinate of the point *A*_2_. Taking the time derivation on both sides of the equal sign of Formula (7), the corresponding relationship between the speed of the end point of the mechanism and the speed of the active joint can be obtained:(8)$$\left[\begin{array}{c} {\dot{x}}_{D}\\ {\dot{y}}_{D} \end{array}\right]=\left[\begin{array}{c} {\dot{d}}_{1}\\ {\dot{d}}_{2} \end{array}\right]-c\dot{\theta }\left[\begin{array}{c} -\sin\theta \\ \cos\theta \end{array}\right]$$
where $\dot{\theta }$ is the angular velocity of the linkage *A*_2_*B*_2_ rotating around the fixed point *R*. By substituting the relation between *θ* and the active joint displacements *d*_1_ and *d*_2_ in Formula (1) into Formula (8), the relation between the velocity of terminal point *D* and the active joint velocity can be obtained:(9)$$\left[\begin{array}{c} {\dot{x}}_{D}\\ {\dot{y}}_{D} \end{array}\right]=J\left[\begin{array}{c} {\dot{d}}_{1}\\ {\dot{d}}_{2} \end{array}\right]$$
where $J=\left[\begin{array}{cc} 1-Ed_{2}sin\theta & E(d_{1}-a)sin\theta \\ Ed_{2}cos\theta & 1-E(d_{1}-a)cos\theta \end{array}\right]$ is the Jacobian matrix of the planar 2-DOF mechanism and $E=\frac{c}{{\left(d_{1}-a\right)}^{2}+d_{2}^{2}}$.

When det(***J***) = 0, we obtain
(10)$$1-\frac{c}{\sqrt{{\left(d_{1}-a\right)}^{2}+d_{2}^{2}}}=0$$

When $c=\sqrt{{\left(d_{1}-a\right)}^{2}+d_{2}^{2}}$, Formula (10) is established. Combining with Figure 3, it can be seen that when the formula is established, point *D* coincides with point *R*, which means that the manipulator based on this mechanism is only in a singular situation when the end of the surgical instrument passes through the incision. It only occurs during the insertion and extraction of the surgical instrument, so it has no practical significance in the actual surgical process.

### 2.4. Kinematic Performance Analysis

The condition number (*k*) of the Jacobian matrix is often used for kinematic performance analysis [14], and its value is the ratio of the largest singular value to the smallest singular value of the Jacobian matrix, namely:(11)$$k=\frac{\sigma_{\max}(J)}{\sigma_{\min}(J)}$$

Obviously, the value range of *k* is [1, +∞]. If its value is 1, the mechanism is isotropic; if its value is +∞, the mechanism is in a singular situation. Therefore, when the value of *k* is smaller, the kinematic performance of the mechanism is better.

Since the value of *k* is determined by the Jacobian matrix, which is a parameter of the joint motion, the value of *k* will also be different when the mechanism is in different configurations. To solve this problem, in order to describe the global motion performance of the mechanism, Gosselin et al. [15] proposed a global condition number index, which is defined as
(12)$$\eta =\frac{\int_{W}(\frac{1}{k})dW}{\int_{W}dW}$$
where *W* is the reachable working space of the organization. Obviously, the range of *η* is (0, 1); the larger the value, the better the kinematic performance of the mechanism.

In this mechanism, mobile joint Ⅰ and mobile joint Ⅱ are selected as the active joints. In the inverse kinematics analysis, the displacements *d*_1_ and *d*_2_ of mobile joint Ⅰ and mobile joint Ⅱ should be obtained according to the position of point *D*. To simplify the analysis, the relationship between the coordinates of point *E* and *d*_1_ and *d*_2_ is obtained first. The coordinate of point *E* can be represented by the coordinate of point *D* as
(13)$$P_{E}=P_{D}-{[b\;0]}^{T}$$

The position vector of point *E* can be expressed as
(14)$$P_{E}=P_{A_{1}}-c\omega$$
where ***P****_E_* = [*x_E_ y_E_*]^T^ is the coordinate of point *E*; ***P***_*A*_1__ = [*d*_1_ − *a d*_2_]^T^ is the coordinate of point *A*_1_; and *c* is the distance between point *A*_1_ and point *E,* which is the same as the distance between point *A*_2_ and point *D.* Substituting each variable into Formula (14), the expressions of *d*_1_ and *d*_2_ can be obtained:(15)$$d_{1}=x_{E}-\frac{cx_{E}}{\sqrt{x_{E}{}^{2}+y_{E}{}^{2}}}+a$$
(16)$$d_{2}=y_{E}-\frac{cy_{E}}{\sqrt{x_{E}{}^{2}+y_{E}{}^{2}}}$$
where *d*_1_ and *d*_2_ are the movement amounts of the two active joints, and Formulas (15) and (16) establish the relationship between the movement amount of the active joints and the coordinates of point *E*. Formula (13) establishes the coordinate relationship between point *D* and point *E*. By substituting Formula (13) into Formulas (15) and (16), the inverse kinematics solution of the 2-DOF mechanism can be obtained:(17)$$d_{1}=x_{D}-\frac{c(x_{D}-b)}{\sqrt{{\left(x_{D}-b\right)}^{2}+y_{D}{}^{2}}}+a-b$$
(18)$$d_{2}=y_{D}-\frac{cy_{D}}{\sqrt{{\left(x_{D}-b\right)}^{2}+y_{D}{}^{2}}}$$

From Formulas (17) and (18), it can be known that for a certain point in the workspace, its coordinate [*x_D_ y_D_*]^T^ is a definite quantity. Then, the values of *d*_1_ − *a* and *d*_2_ are related to the structural parameters *b* (the length of linkage *A*_1_*A*_2_) and *c* (the length of linkage *A*_1_*E*). By substituting it into the Jacobian matrix (***J***) of the mechanism, it can be found that the Jacobian matrix is only related to *c*, and *b* is eliminated. Figure 3 shows that size parameter *b* only affects the fixed x-coordinate of point *R* and has no effect on the kinematic performance of the mechanism. To sum up, only the value of size parameter *c* affects the Jacobian matrix.

Figure 4a shows the change in *η* with *c*. It can be seen that *η* first increases and then decreases with the increase in *c* and reaches the maximum value of 0.64 when *c* = 230 mm. In order to study the specific influence of *c* on *η* in the working space, the distribution diagram of the reciprocal of the condition number (*k*) in the working space is drawn when the value of *c* is 230 mm, as shown in Figure 4b. Although the kinematic performance of the mechanism is optimal when *c* = 230 mm, the value of the specific dimension *c* also needs to consider some limitations in the structural design.

After the investigation, the top angle of the conical motion space for surgical instrument movement was 60°, and the insertion depth range was 50–200 mm. The limitation of the depth range means that the maximum distance from the terminal point *A*_2_ to the fixed point *R* is 200 mm; that is, the minimum value of dimension *c* is 200 mm. It is easy to see in Figure 3 that *a* should meet
(19)a≥sin(60°/2)×(c−50)

It can be seen in Figure 4a that the kinematic performance of the mechanism is optimal when the length parameter *c* is 230 mm. However, when designing the manipulator with this parameter, due to the existence of some structural interference, such as that the range of motion of mobile joint I and mobile joint II cannot start from 0, its workspace is not necessarily larger than the required workspace. In the design of the mechanical arm, parameter *c* is taken as 250 mm, and parameter *a* is taken as 120 mm according to Formula (19). Under this condition, the ranges of motion of mobile joint I and mobile joint II are 25–215 mm and 35–200 mm.

Considering the construction of hardware, the final design results are shown in Table 1 and Table 2:

## 3. Kinematic Analysis

### 3.1. Forward Kinematics Analysis

In order to explain the kinematics of the manipulator, we use the DH algorithm for modeling. Since the manipulator based on this mechanism is not a simple serial or parallel manipulator, the following simplification needs to be made: the 2-DOF RCM mechanism from this plane is simplified into a revolute joint and a mobile joint. Finally, the coordinate system shown in Figure 5 is established according to the D-H method, and the dotted line represents the initial configuration. A coordinate system {0} is established at the first revolute joint, which is the base coordinate system fixed to the base and does not move with the manipulator. A coordinate system {1} is established at the RCM point and represents the revolute joint action of the RCM mechanism at this point. A coordinate system {2} is established at the RCM point to represent the action of the mobile joint realized by the RCM mechanism at this point. Finally, the tool coordinate system {H} is established at the end of the instrument.

For the coordinate system established in Figure 5, according to the definitions of parameters in the D-H method, the D-H parameters of the manipulator are obtained, as shown in Table 3, where *θ*_1_, *θ*_2_, and *d* are joint variables and *L* is the structure size.

The relative positional relationship between each two sets of coordinate systems in the above table can be represented by the transformation matrix *^n^**T**_n_*_+1_:(20)Tn+1n=Cθn+1−Sθn+1Cαn+1Sθn+1Sαn+11an+1Cθn+1Sθn+1Cθn+1Cαn+1−Cθn+1Sαn+1an+1Sθn+10Sαn+1Cαn+1dn+10001

By substituting the parameters in the table into Formula (20) and multiplying the transformation matrix, the forward kinematics equation of the manipulator is obtained, as shown in Formula (21):(21)TH0=T10×T21×TH2=−Cθ1Sθ2Sθ1−Cθ1Cθ2−dCθ1Cθ2−Sθ1Sθ2Cθ1−Sθ1Cθ2−dSθ1Cθ2Cθ20Sθ2dSθ2+L0001
where $C\theta_{1}=\cos\theta_{1}$, $S\theta_{1}=\sin\theta_{1}$, $C\theta_{2}=\cos\theta_{2}$, and $S\theta_{2}=\sin\theta_{2}$.

Formula (21) contains the position and attitude information of the coordinate system {H} in the base coordinate system {0}, which can be converted as follows:(22)TH0=RH0PH001

The rotation transformation (RH0) and translation transformation (PH0) are
(23)RH0=−Cθ1Sθ2Sθ1−Cθ1Cθ2−Sθ1Sθ2Cθ1−Sθ1Cθ2Cθ20Sθ2
(24)PH0=−dCθ1Cθ2−dSθ1Cθ2dSθ2+L1

From Figure 3, the relationship between *θ*_2_, *d, d*_1_, and *d*_2_ can be obtained as
(25)$$d=c-\sqrt{{\left(d_{1}-a\right)}^{2}+d_{2}{}^{2}}$$
(26)$$\theta_{2}=\arctan(\frac{a-d_{1}}{d_{2}})$$

By substituting Formulas (25) and (26) into Formula (24), the forward kinematics solution of the manipulator is obtained:(27)PH0=−(c−(d1−a)2+d22)Cθ1C(arctan(a−d1d2))−(c−(d1−a)2+d22)Sθ1C(arctan(a−d1d2))(c−(d1−a)2+d22)S(arctan(a−d1d2))+L1

### 3.2. Analysis of the Motion Space of the Manipulator

This paper uses the Monte Carlo method to analyze the working space of the designed manipulator. First, according to the key rod size and the active joint motion range shown in Table 1 and Table 2, the random number function is used in MATLAB to randomly generate the motion amount within the motion range of each active joint, and then the motion amount of each active joint is arranged and combined. Finally, the coordinate value of the end of the surgical instrument is obtained by solving the forward kinematics formula shown in Formula (24). A large number of coordinate values are generated in the point cloud space, which is the reachable movement space of the manipulator. Programming in MATLAB software, the workspace shown in Figure 6 and the projection of the workspace on the plane of each coordinate system are obtained. For the convenience of display, the base coordinate system {0} in Figure 5 is translated along the Z axis to make its origin overlap with the RCM point, and the distribution image of the workspace is drawn in the translated coordinate system. The above simulation shows that the designed manipulator can meet the workspace requirements.

### 3.3. Inverse Kinematics Analysis

From the forward kinematics analysis in Section 3.1, it can be seen that the manipulator is a 3-DOF manipulator. The position and attitude of the tool coordinate system {H} are not independent, and its position is the input of the manipulator control. The position of the end point (that is, the origin of the tool coordinate {H}) is described in the base coordinate system {0} as
(28)PH0=px0py0pz01T

Then, from forward kinematics Formula (27), we can obtain:(29)px0py0pz01T=−(c−(d1−a)2+d22)Cθ1C(arctan(a−d1d2))−(c−(d1−a)2+d22)Sθ1C(arctan(a−d1d2))(c−(d1−a)2+d22)S(arctan(a−d1d2))+L1

The inverse kinematics solution of the manipulator can be obtained:(30)$$\theta_{1}=\tan^{-1}\frac{y}{x}$$
(31)$$d_{1}=a-(c-d)\frac{z-L}{d}$$
(32)$$d_{2}=(c-d)\frac{\sqrt{x^{2}+y^{2}}}{d}$$
where $d=\sqrt{x^{2}+y^{2}+{\left(z-L\right)}^{2}}$, *a*, *c*, and *L* are structural dimension parameters.

It can be seen in Table 2 that the motion range of the revolute joint satisfies −38° < *θ*_1_ < 38°, so the value of *θ*_1_ can be directly obtained from Formula (30), thus avoiding the situation of multiple solutions.

## 4. Prototyping

According to the above design, the physical prototype shown in Figure 7 was finally obtained. Most of the prototype parts were machined from 6061 alloy, and the overall structural part of the prototype weighed about 2.5 kg, which had the advantages of small size and light weight. For the convenience of experimental operation, the surgical instrument in Figure 7 adopted grasping forceps. The instrument was driven by a micro screw motor to control the opening and closing.

### 4.1. Master–Slave Control

In order to further verify the feasibility of this newly designed RCM mechanism manipulator, we used the master–slave heterogeneous method to carry out master–slave control, and the selected master manipulator was Geomagic’s master manipulator Touch, as shown in Figure 8.

Considering the characteristics of minimally invasive surgery, a consistent motion control strategy was adopted in the master–slave motion control strategy to realize the point-to-point motion mapping between the handle of the master manipulator and the end of the instrument. In the master–slave coordinate system definition shown in Figure 9, the origin motion direction of the master manipulator handle motion coordinate system (*H-xyz*) is consistent with the origin motion direction of the surgical instrument end motion coordinate system (*T-xyz*) in the geodetic coordinate system *G-xyz* description. Due to the mismatch of the working space caused by the different parameters, such as the joint configuration of the master and slave ends and the length of the linkage, an incremental motion control strategy needed to be adopted, which could cut off the master–slave mapping at any time, adjust the position of the master manipulator, and ensure that the operator operated in a comfortable space. In addition, in order to meet the requirements of fine operation during the operation, the master–slave control strategy needed to add stroke-proportional control to improve the quality of the operation. Combining the above-mentioned consistent motion control, incremental motion control, and stroke-proportional control, the master–slave follow-up strategy of the minimally invasive surgical manipulator could be realized. The specific algorithm is shown in Figure 10.

In the master manipulator part, the position and pose parameters in reference frame *M-xyz* were obtained by calculating the joint parameter ***q****_m_* through the forward kinematics transformation formula of the master manipulator. The parameters were collected before and after every fixed time, and the difference was used to obtain the increment r˙HM of the master manipulator terminal in *M-xyz*. In the master–slave control system, through the aforementioned consistent motion control, coordinate r˙HM was transformed and then scaled according to the master–slave mapping ratio to obtain the increment r˙TS of the end of the manipulator in its reference coordinate system *S-xyz*. This was added it to the actual manipulator end pose parameter (rT0S) to obtain the desired pose (rTS) and finally obtain the desired joint angle (***q****_s_)* by calculating (from the inverse kinematics of the manipulator) the actual pose parameter (rT0S) of the slave manipulator, which was calculated from the actual joint angle (${q^{\prime }}_{s}$) of the slave manipulator through the forward kinematics transformation formula of the slave manipulator. In the slave manipulator part, a typical three-closed-loop motion control system of current, speed, and position was used to control the position of the manipulator’s joint motors.

According to the master–slave following algorithm described in Figure 10, the specific formula can be described as
(33)rTs=k×R−1SG×RMG×r˙HM+rT0s

In Formula (34), the actual joint angle (${q^{\prime }}_{s}$) is substituted into the forward kinematics transformation formula of the manipulator to obtain the actual manipulator end pose parameter (rT0s). After the desired pose (rTS) of the robotic arm was obtained, the desired joint angle ($q_{s}$) could be obtained by substituting the inverse kinematics Formulas (29)–(31) derived in Section 3.3 for calculation.
(34)rT0S=TH0×PH=−Cθ1Sθ2Sθ1−Cθ1Cθ2−dCθ1Cθ2−Sθ1Sθ2Cθ1−Sθ1Cθ2−dSθ1Cθ2Cθ20Sθ2dSθ2+L0001×0001

### 4.2. Master–Slave Control Experiment

#### 4.2.1. Trajectory Tracking Experiment

In the experiment, the master manipulator was operated to control the slave manipulator so that its ends reached the four target positions on the graph paper in turn. The four target positions were the four vertices of a 50 mm × 50 mm rectangle. During the experiment, the master–slave mapping ratio was selected as 2:1. The experimental process is shown in Figure 11.

The master manipulator position data could be directly read by the host computer, and its trajectory during the experiment is shown in Figure 12a. In this paper, the collected joint motor encoder data were used to calculate the position information of the end of the manipulator through the forward kinematics formula, and its trajectory is shown in Figure 12b. It can be seen in Figure 12 that the side length of the rectangular track at the master manipulator end was twice the length of the rectangle at the slave manipular end, which corresponded to the master–slave mapping ratio of 2:1 in the experimental setup.

During the experiment, in order to pursue the best operation posture, when the main hand movement was not suitable for the operation space, the master manipulator adjustment button was pressed, the main hand was removed to a suitable position, and the experimental operation continued. The collected motion information of the master–slave manipulator end position in the *xyz* directions is shown in Figure 13. As can be seen in the figure, the position of the master manipulator in the *xyz* directions changed abruptly at about 12 s, which was the adjustment of the master manipulator at this time. The position of the end of the slave manipulator did not change abruptly, and the movement trend of the master and slave ends was the same. Due to the different definitions of the reference coordinate systems of the master and slave ends, the position information of the master and slave manipulator were represented in their respective reference coordinate systems. Therefore, in Figure 13, the x-direction trajectory of the master manipulator end in its reference system corresponds to the z-direction trajectory of the manipulator, the y-direction trajectory of the master manipulator end corresponds to the x-direction trajectory of the manipulator, and the z-direction trajectory of the master manipulator end corresponds to the y-direction trajectory of the manipulator.

#### 4.2.2. Minimally Invasive Surgical Operation Simulation Experiment

The experimental setup is shown in Figure 14. A transparent acrylic cone was used to simulate the surgical operation space. A small hole with a diameter of 12 mm was set at the top of the cone to simulate a surgical incision. The diameter of the ball was 2.5 mm, the height of the column where the ball was placed in the inner ring was 10 mm, and the height of the outer column was 15 mm. In the experiment, the operator needed to hold the master manipulator, use the master–slave mapping ratio of 3:1, and operate the manipulator to grip and transport the ball. The experimental process is shown in Figure 15.

During the experiment, the trajectory data of the master manipulator and the encoder data of the joint motor of the manipulator were collected. According to the master–slave control algorithm, the expected trajectory of the slave hand could be obtained, and the actual trajectory of the end of the manipulator could be obtained from the encoder data of the joint motor of the manipulator through a forward kinematics calculation. As shown in Figure 16, the slave hand could follow the desired trajectory of the slave hand well and complete the expected operation of the operator. The trajectory errors of the manipulator in all directions during the operation are shown in Figure 17.

The maximum error and average error in the master–slave following process are shown in Table 4. It can be seen in the table that the maximum value of the manipulator following error during the operation was 1.49 mm and the average error was 0.45 mm, among which the proportion of error less than 0.5 mm was 62.54% and the proportion of error less than 1 mm was 96.54%. David and Kwartowitz [16] determined through experiments that the positioning error of the Da Vinci surgical robot system was no more than 1.2 mm, which was similar to the positioning accuracy of the surgical robot in this paper. Considering that the operator can always make active adjustments through visual feedback to move the surgical instrument to the target position, the errors shown in Table 4 will not affect the manipulative performance of the manipulator.

Figure 18 shows the following curves of the three active joints of the manipulator in the master–slave motion. The time period when the joints changed sharply was selected for enlargement. It can be seen in the figure that the delay of each joint was basically 60–80 ms, and the maximum delay was less than 100 ms. The delay included the time consumed by the communication and the master–slave mapping calculation, and the rest was the position following delay from the robotic arm to the master manipulator. According to the experiment by Marescaux et al., the acceptable delay limit for doctors for safe operation is 330 ms [17], so the delay of the robotic arm can meet the needs of use and will not affect the fluency of the doctor’s operation.

The diameter of the surgical instrument rod used in the experiment was 5 mm. Before the experiment, the axis of the surgical instrument passed through the center of the 12 mm diameter hole at the top of the acrylic cone. During the operation, the surgical instrument produced pitch and yaw motions, but it did not contact the wall of the small hole at the top of the acrylic cone, which proved the existence of the fixed point of the mechanical arm, and the deviation during the movement did not exceed 2.5 mm, indicating that the robotic arm can meet the surgical requirements that the fixed point offset cannot exceed 5 mm.

## 5. Conclusions

This project aimed to design a minimally invasive surgical manipulator and realize its master–slave control. On this premise, a new planar 2-DOF RCM mechanism was designed. Based on this mechanism, a prototype of the minimally invasive surgical manipulator was built. The arm could achieve RCM motion, and this was mathematically proven. In this paper, the singularity and kinematic performance of the mechanism were analyzed using the Jacobian matrix, and the global condition number index reached its maximum when the length of linkage *c* was 230 mm and its value was 0.64. In this paper, the motion space of the proposed RCM mechanism was analyzed. Regarding the main structural parameters, *c* was 250 mm, *a* was 120 mm, and the range of motion of mobile joint I and mobile joint II were 25 mm and 215 mm, and 35 mm and 200 mm, respectively. Based on this parameter, the active joint design of the manipulator and the selection of the joint motor were carried out. After that, the three-dimensional model of the manipulator was established, and the physical prototype was obtained by processing and assembling. Finally, the kinematics model of the manipulator was established using the D-H method. We used a master–slave experiment to test the mechanical properties of the manipulator. First, through the trajectory tracking experiment, it was proven that the robotic arm could track the movement of the master manipulator smoothly, which preliminarily showed that the robotic arm and the master–slave control system could basically meet the application requirements of minimally invasive surgery. Second, the remote operation gripping and handling experiment of the manipulator was carried out. In the experiment, the average following error of the end of the manipulator was 0.45 mm, and the maximum following error was 1.49 mm, which was similar to the Da Vinci surgical robot. Under the condition that the master–slave mapping ratio was 3:1, the master–slave following delay was between 60 and 80 ms, which would not affect the smoothness of an operation. Overall, the manipulator can meet the application requirements.

## Figures and Tables

**Figure 1 sensors-23-02361-f001:**
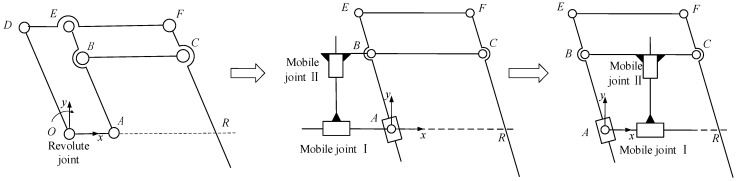
Deduction of planar 2-DOF RCM mechanism.

**Figure 2 sensors-23-02361-f002:**
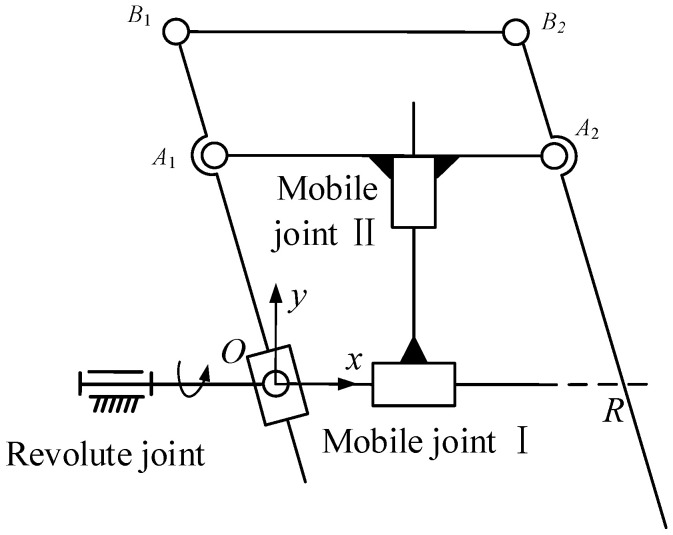
A 3-DOF RCM mechanism.

**Figure 3 sensors-23-02361-f003:**
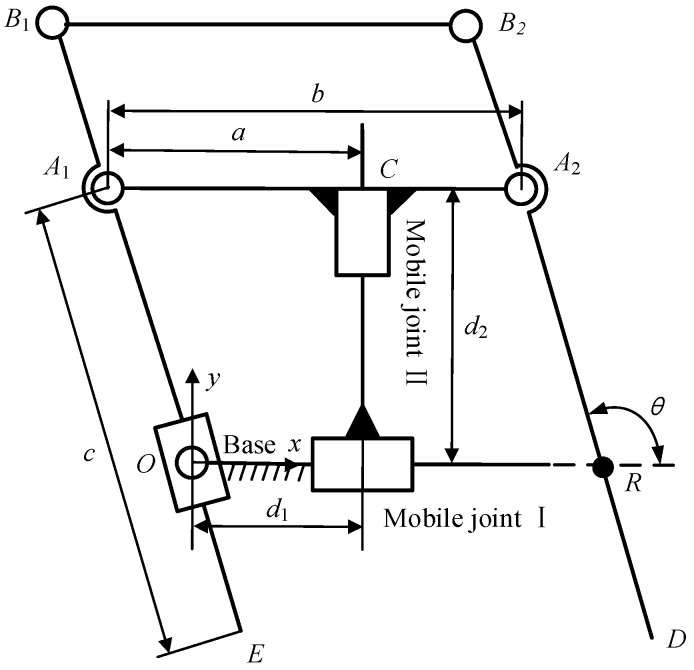
Dimensioning for kinematic analysis.

**Figure 4 sensors-23-02361-f004:**
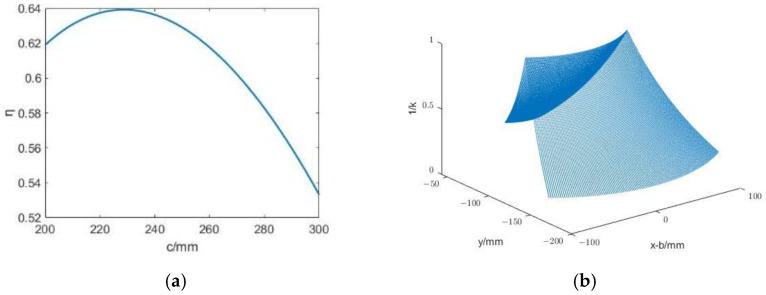
Mechanism motion performance index. (**a**) *η* varies with *c*. (**b**) The distributions of 1/k when *c* = 230 mm.

**Figure 5 sensors-23-02361-f005:**
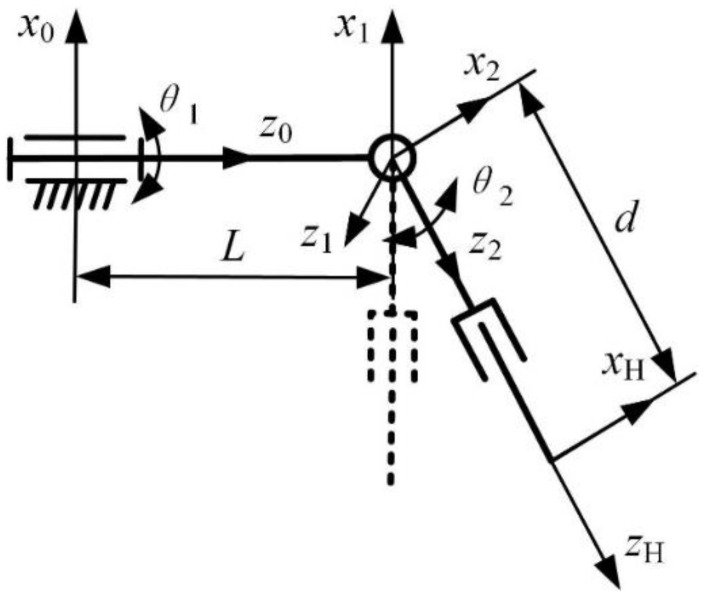
D-H coordinate system of surgical robotic arm.

**Figure 6 sensors-23-02361-f006:**
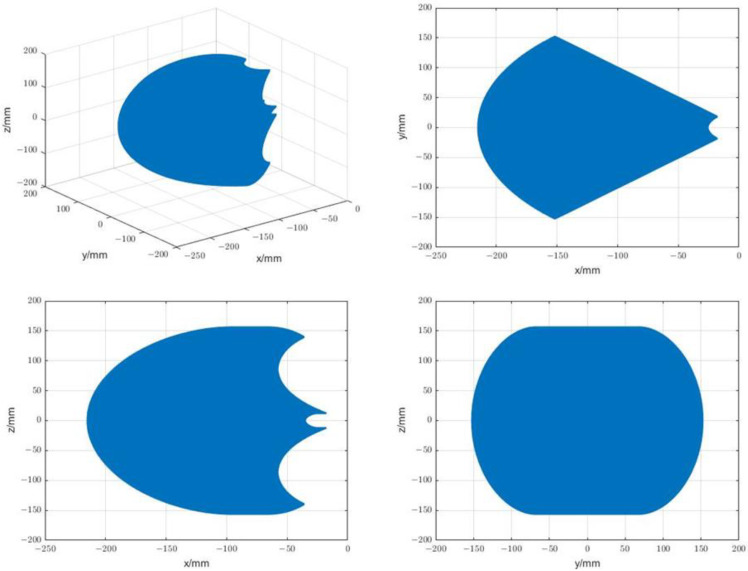
The 3D workspace and 3 plane projections.

**Figure 7 sensors-23-02361-f007:**
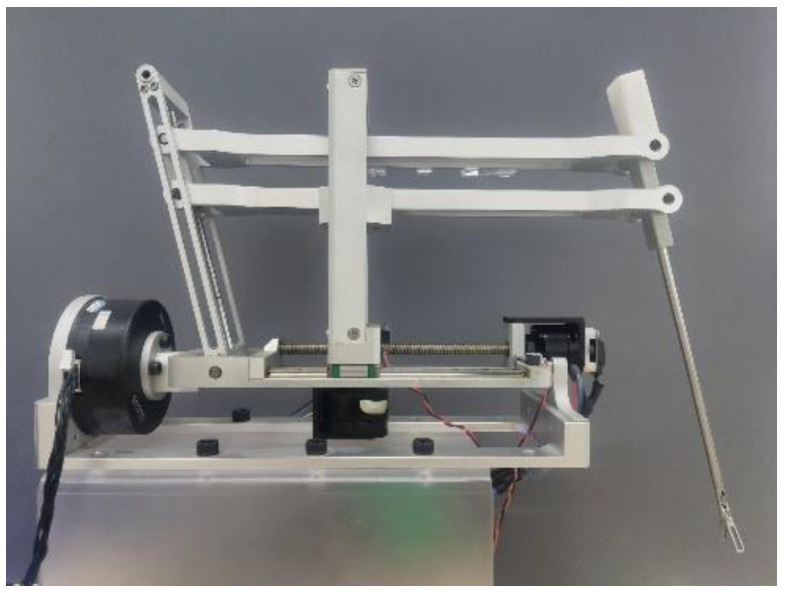
Prototype.

**Figure 8 sensors-23-02361-f008:**
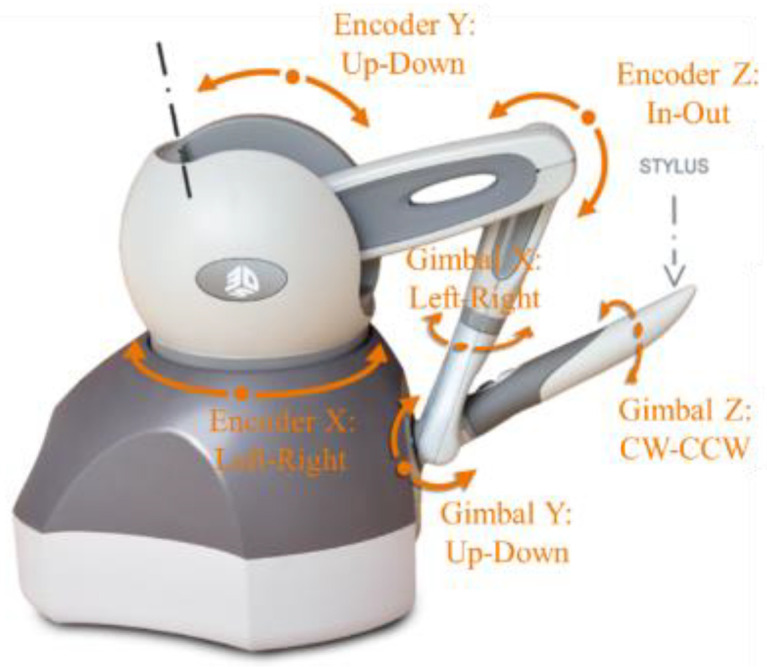
Master manipulator Touch.

**Figure 9 sensors-23-02361-f009:**
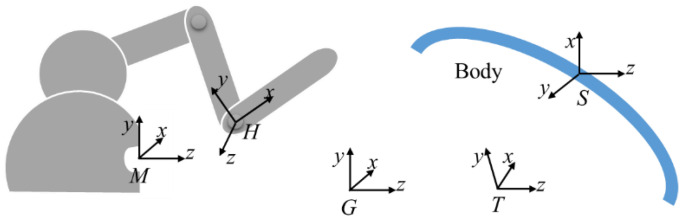
Master–slave coordinate system definition.

**Figure 10 sensors-23-02361-f010:**
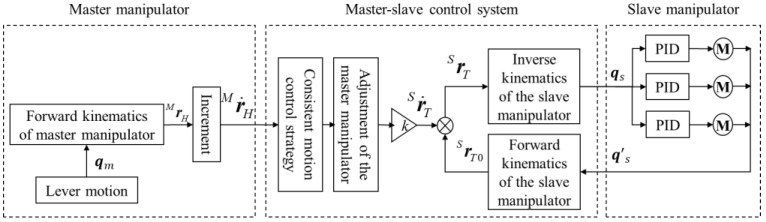
Master–slave control algorithm flow.

**Figure 11 sensors-23-02361-f011:**
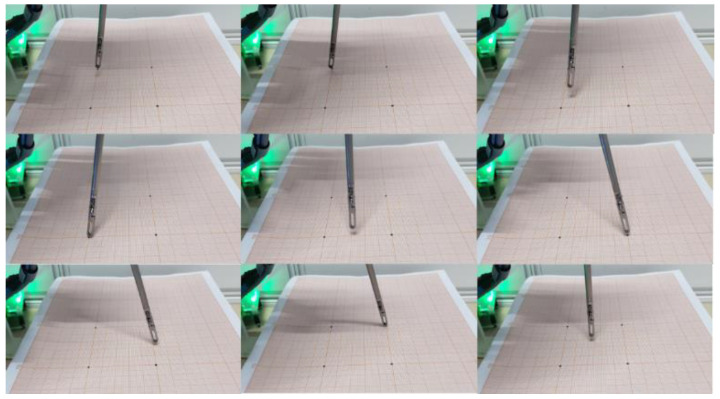
Tracking experiment process.

**Figure 12 sensors-23-02361-f012:**
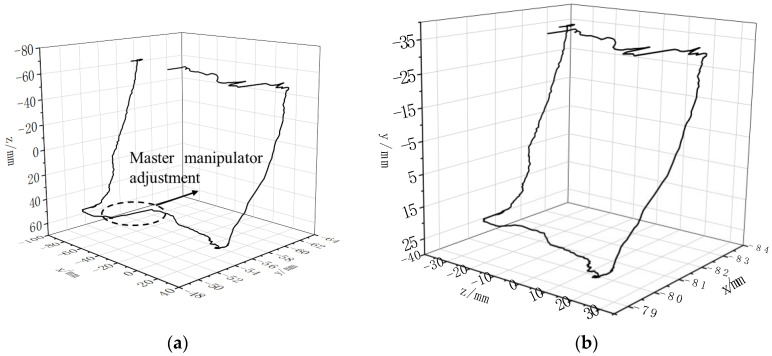
Master–slave trajectory. (**a**) Master manipulator trajectory. (**b**) End trajectory of the slave manipulator.

**Figure 13 sensors-23-02361-f013:**
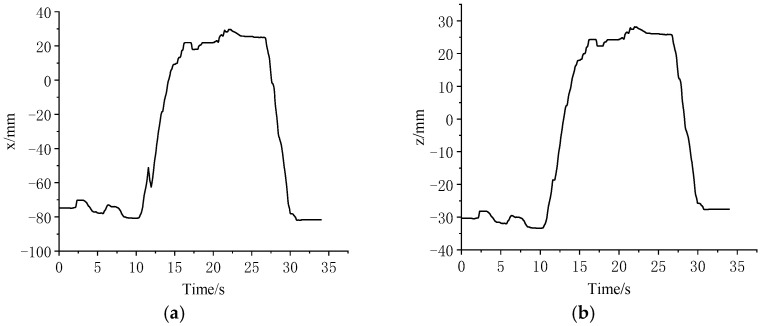
Trajectories of master and slave in the *xyz* directions. (**a**) Master x direction. (**b**) Slave z direction. (**c**) Master y direction. (**d**) Slave x direction. (**e**) Master z direction. (**f**) Slave y direction.

**Figure 14 sensors-23-02361-f014:**
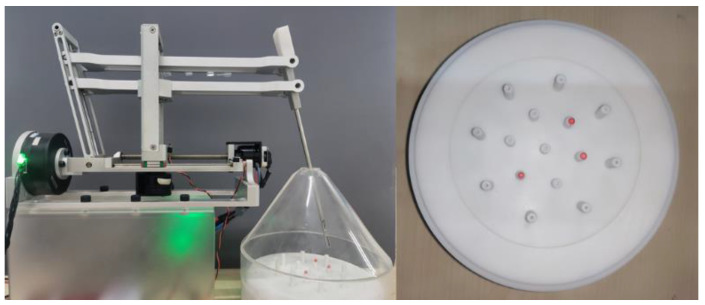
The experimental setup of the robotic arm gripping and handling.

**Figure 15 sensors-23-02361-f015:**
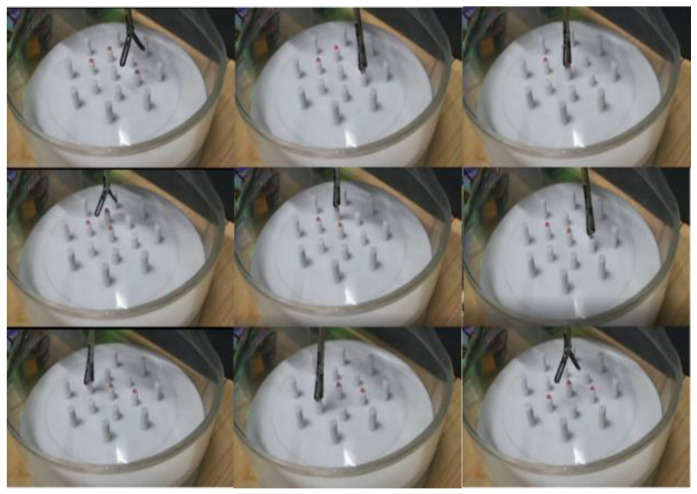
The experimental process of gripping and handling.

**Figure 16 sensors-23-02361-f016:**
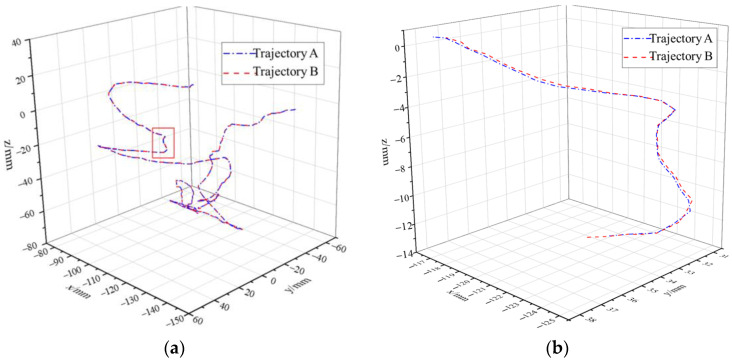
Master–slave following trajectory. (**a**) The tracking effect at the end of the manipulator. (**b**) Enlargement of the tracking effect at the end of the manipulator.

**Figure 17 sensors-23-02361-f017:**
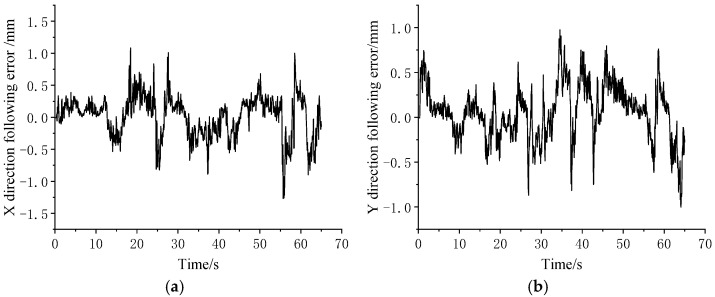
Master–slave following error in gripping and handling experiments. (**a**) X direction. (**b**) Y direction. (**c**) Z direction. (**d**) Resultant.

**Figure 18 sensors-23-02361-f018:**
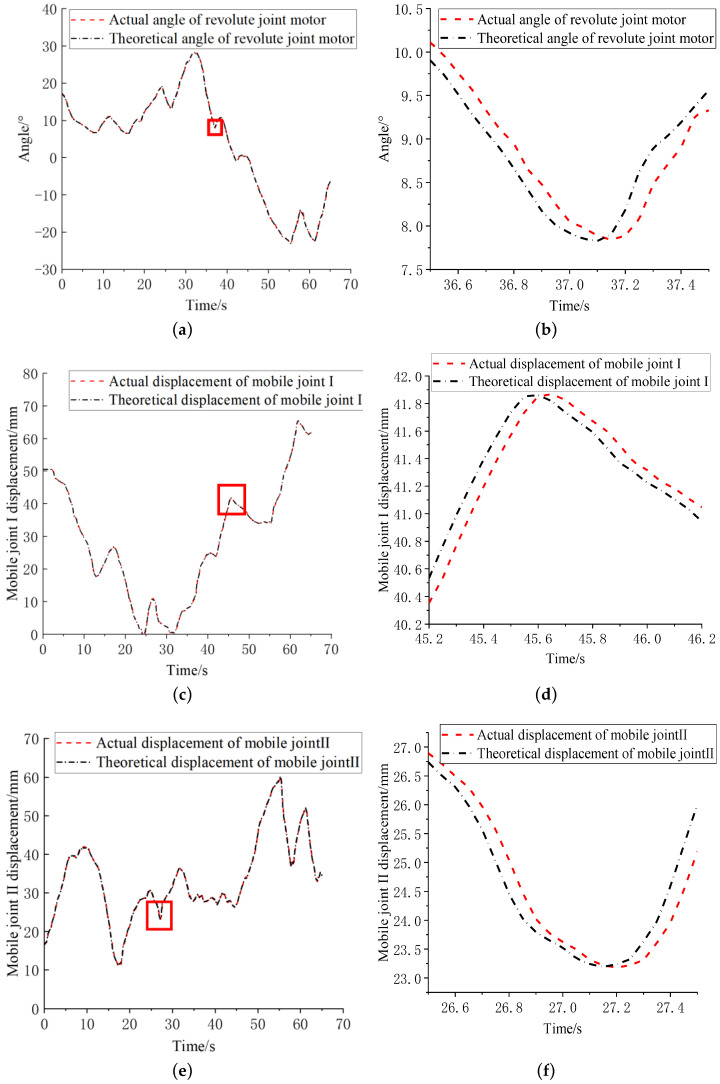
Robotic arm joint tracking effect where zoom in takes place in red boxes. (**a**) Revolute joint tracking effect. (**b**) Enlargement of revolute joint tracking effect. (**c**) Mobile joint I tracking effect. (**d**) Enlargement of mobile joint I tracking effect. (**e**) Mobile joint II tracking effect. (**f**) Enlargement of mobile joint II tracking effect.

**Table 1 sensors-23-02361-t001:** Key linkage length.

Parameter	*A* _1_ *C = a*	*A*_1_*A*_2_ = *b*	*A*_1_*E* = *c*
Length (mm)	120 mm	260 mm	250 mm

**Table 2 sensors-23-02361-t002:** Range of the active joints.

Active Joint Variables	Revolute Joint *θ*_1_	Mobile Joint Ⅰ *d*_1_	Mobile Joint Ⅱ *d*_2_
Range of motion	−38–38°	22–215 mm	35–200 mm

**Table 3 sensors-23-02361-t003:** D-H parameters of surgical manipulator.

#	Joint Angle *θ*	Joint Offset *d*	Linkage Length *a*	Linkage Torsion Angle *α*
0-1	*θ* _1_	*L*	0	−90°
1-2	−90° + *θ*_2_	0	0	90°
2-H	0	*d*	0	0

**Table 4 sensors-23-02361-t004:** Master–slave following errors.

	X Direction	Y Direction	Z Direction	3D Space
**Maximum Error (mm)**	−1.27	−1.00	1.13	1.49
**Average Error (mm)**	0.25	0.24	0.17	0.45

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
