# Peer review of "Design and Performance Verification of a Novel RCM Mechanism for a Minimally Invasive Surgical Robot"

_sensors, 2023, doi:10.3390/s23042361_

Round 1

Reviewer 1 Report

This paper presents a planar RCM mechanism for minimally invasive surgical robots. It is introduced that this mechanism can complete the required motion for surgery without adding a mobile unit. The topic is interesting and important for practical surgery application. This paper is well organized and should be published on Sensors.

Major:

It is a little wordy in terms of content, and it is suggested to highlight key problems and solutions. For example, the figures and formulas should be combined, and the results can be analyzed more concise.

Minor:

1)      the complete spelling of RCM should be given in the first time.

2)      degrees of freedom should be abbreviated.

3)      Error figure citing shows frequently, please check.

4)      Since there are a lot of variables involved, it is suggested to add a symbol list. E.g. Parameter description is repeated, such as “where: d1and d2 are the displacements of mobile joint I and mobile joint II”.

Author Response

Dear Reviewer:
I am very grateful to your comments for the manuscript. Those comments are all valuable and very helpful for revising and improving our paper, as well as the important guiding significance to our researches.  We have studied comments carefully and have made correction which we hope meet with approval.  The main corrections in the paper and the responds to the reviewer’s comments are as flowing:
Response to major comment: Thank you for this very insightful comment. We have made some changes in the content. There are a lot of pictures and formulas in the article. We think that the combination of figures and formulas cannot clearly describe the problem, so we choose to express them separately.
Response to minor comment 1: We are very sorry for our negligence. We have made correction according to your comments.
Response to minor comment 2: We are very sorry for our negligence. We have made correction according to your comments.
Response to minor comment 3: After careful inspection, we found no such problem in the manuscript we provided. This is a problem caused by your editor's re-typesetting. I'm sorry.
Response to minor comment 4: In the figure, we have marked the length of each connecting rod, and made reasonable explanations in the first appearance of parameters.
We tried our best to improve the manuscript and made some changes in the manuscript.  These changes will not influence the content and framework of the paper.  We appreciate for Editors/Reviewers’ warm work earnestly, and hope that the correction will meet with approval. Once again, thank you very much for your comments and suggestions.

Reviewer 2 Report

The article proposes a new 2-degree-of-freedom RCM mechanism, this mechanism allows for both the pitching and translational movements required for surgery, without the need for an additional movable joint at the end, the risk of interference with surgical machinery is better avoided. The article describes the design process of the institution clearly, mathematical proof that the mechanism can achieve RCM motion, and analyzed the singularity and motion performance of the mechanism, after that, the key linkage length and the range of movable joints were designed. Applying the D-H method, kinematic model of the robot arm was established. Machining out the robot arm prototype, trajectory tracking experiments were conducted using master-slave control, and carried out gripping and handling experiments, the experimental results show that the positioning accuracy of the manipulator is good and can meet the requirements for the use of surgery.

Based on the content of the article, the following revisions to the article are proposed:

1.The description of the picture and the position shown in the formula is incorrect several times in this article, all show no reference source found, please double-check the description of the image and formula position in the full text, such errors are checked and carefully corrected. For example, page 2, penultimate line 8, page 5, penultimate line 4, etc.

2. Article fourth page position line 10, the “A1B2” in “ω=[cosθ sinθ]T is the direction vector of the straight line A1B2” refers to the wrong person, should it be amended to A1B1.

3. Page 6, line 5 of the article, the reference number is not written correctly.

4. The article mentions substituting the formula into the Jacobi matrix J in line 3 on page 7, the Jacobi matrix is only related to the parameter c, and parameter b is eliminated, please list the results of the Jacobi matrix J after substituting the formula, or explain in detail how parameter b has been eliminated.

5. How the key linkage lengths in Table 1 and the range of movable joints in Table 2 on page 7 of the article were determined, Please add relevant content to provide an explanation.

6. The z-coordinate values in Figure 12(b) should be located inside the chart, please modify this image

7. Figure 17 is repeated on pages 16-17 of the article, Error in drawing number labelling for X direction, please make changes to the markings in the diagram and to the order in which the images are laid out.

Author Response

Dear Reviewer:
I am very grateful to your comments for the manuscript. Those comments are all valuable and very helpful for revising and improving our paper, as well as the important guiding significance to our researches.  We have studied comments carefully and have made correction which we hope meet with approval.  The main corrections in the paper and the responds to the reviewer’s comments are as flowing:
Response to comment 1: After careful inspection, we found no such problem in the manuscript we provided. This is a problem caused by your editor's re-typesetting. I'm sorry.
Response to comment 2: We are very sorry for our negligence. We have made correction according to your comments.
Response to comment 3: After careful inspection, we found no such problem in the manuscript we provided. This is a problem caused by your editor's re-typesetting. I'm sorry.
Response to comment 4: Take the derivative on both sides of formula 7. Because b is a constant, the Jacobian matrix does not contain b. It can also be seen from Figure 2-3 that b only affects the x-coordinate of the fixed-point R and has no effect on the kinematic performance of the mechanism.
Response to comment 5: We have made correction according to the Reviewer’s comments.
Response to comment 6: We have made correction according to the Reviewer’s comments.
Response to comment 7: We are very sorry for our negligence. We have made correction according to your comments.
We tried our best to improve the manuscript and made some changes in the manuscript.  These changes will not influence the content and framework of the paper.  We appreciate for Editors/Reviewers’ warm work earnestly, and hope that the correction will meet with approval. Once again, thank you very much for your comments and suggestions.

Reviewer 3 Report

1. This paper proposes a structure that can realize the translation of surgical instruments, which can be described more specifically when it is first mentioned, such as which axis to translate in which direction, which can make the description more clear.

2. At various points in the article "Error! Reference source not found. "The author can check to see if there is a problem with formatting insertion, which is causing some of the text's fluency to be blocked.

3. In the derivation of the formula P on page 4, there is a description error, the meaning of ωshould be for A2B2

4. The marking format of references needs to be unified, and the phenomenon of mixed use appears in the 13th and 14th

5. In the chapter Kinematic Performance Analysis, the author uses global condition number index to describe kinematic performance. What are the reasons for such choice? How to explain the influence of the degree of anisotropy fluctuation in the working space on motion performance

6. There are obvious differences between the end trajectories of (a) and (b) in Master-slave trajectory. If it is caused by the picture Angle, I hope the author can unify the perspective. If not, please explain the reasons

Author Response

Dear Reviewer:
I am very grateful to your comments for the manuscript. Those comments are all valuable and very helpful for revising and improving our paper, as well as the important guiding significance to our researches.  We have studied comments carefully and have made correction which we hope meet with approval.  The main corrections in the paper and the responds to the reviewer’s comments are as flowing:

Response to comment 1: The degree of freedom of movement of the surgical instrument itself needs to add a moving joint at the end. The mechanism in this paper can directly realize the movement of the surgical instrument without moving the joint, and the direction of realization is the three-dimensional direction required by the operation. The three-dimensional space reached is described in Section 3.2.

Response to comment 2: After careful inspection, we found no such problem in the manuscript we provided. This is a problem caused by your editor's re-typesetting. I'm sorry.

Response to comment 3: We are very sorry for our negligence. We have made correction according to your comments.

Response to comment 4: After careful inspection, we found no such problem in the manuscript we provided. This is a problem caused by your editor's re-typesetting. I'm sorry.

Response to comment 5: References 14 and 15 are cited to explain that, in brief, for parallel mechanisms, isotropy means that the transfer characteristics of their motion do not change with different directions, and the evaluation index of dynamic isotropy can be expressed by formula (12).

Response to comment 6: The angle is uniform. From Figure 4-6, we can see that the shape of the end track is the same. The difference is that the coordinates of the two pictures are different, because the ratio of master-slave mapping is 2:1.As can be seen from the figure, the position of the master manipulator in the xyz direction changes abruptly at about 12s, which is the adjustment of the master manipulator at this time.The position of the end of the slave manipulator does not change abruptly, and the movement trend of the master and slave ends is the same.

We tried our best to improve the manuscript and made some changes in the manuscript.  These changes will not influence the content and framework of the paper.  We appreciate for Editors/Reviewers’ warm work earnestly, and hope that the correction will meet with approval. Once again, thank you very much for your comments and suggestions.

Round 2

Reviewer 3 Report

I think this paper can be accept.